# Degradation of Polymer Films of Sodium Alginate during Prolonged Irradiation with X-ray under Ultra-High Vacuum

**DOI:** 10.3390/polym16142072

**Published:** 2024-07-20

**Authors:** Eugene N. Kabachkov, Sergey A. Baskakov, Yury M. Shulga

**Affiliations:** 1Federal Research Center of Problems of Chemical Physics and Medicinal Chemistry, Russian Academy of Sciences, Academician Semenov Avenue 1, 142432 Chernogolovka, Russia; baskakov@icp.ac.ru; 2Osipyan Institute of Solid State Physics RAS, Russian Academy of Sciences, 2 Academician Osipyan Str., 142432 Chernogolovka, Russia

**Keywords:** sodium alginate, degradation, X-ray, irradiation, IR, time-dependent XPS

## Abstract

Sodium alginate (NaAlg) is widely used as a food additive. To study the effect of irradiation with X-ray quanta with energies of 1253.6 eV and 1486.6 eV on the composition of NaAlg, thick films with a smooth surface were prepared, which did not differ in IR spectra from the original powders. The films were irradiated in a high vacuum (3 × 10^−10^ mbar) in the chamber of a Specs PHOIBOS 150 MCD9 XPS spectrometer with an X-ray source power of 150 W and an irradiation duration of up to 300 min, which significantly exceeded the time required to obtain an XPS spectrum. This made it possible to use XPS to monitor changes in the composition of the NaAlg surface directly during irradiation. As a result of the research, it has been established that NaAlg degrades with prolonged irradiation, which is accompanied by a significant decrease in the O/C ratio. When analyzing the dependence of the intensities of individual peaks in the C1s spectrum on the irradiation time, it was found that after 100 min of irradiation, a peak due to the carbonate group appears in the spectrum. The decomposition was also accompanied by a change in the color of NaAlg from white to yellow-brown. In the IR spectrum of the NaAlg film irradiated for 300 min, an absorption band was detected at 1910 cm^−1^, which is usually associated with the presence of allene groups.

## 1. Introduction

Alginates are salts of alginic acid. More than 200 types of alginates are commercially available. The total production of alginates in 2021 exceeded 44,000 tons. Most alginates are extracted from brown seaweed. The applied significance of alginic acid and its derivatives is determined by its structure. This is a high molecular weight polysaccharide, which is a block copolymer of D-mannuronic and L-guluronic acids. Alginates have proven to be biodegradable, renewable, biocompatible, non-immunogenic materials that currently have an immense number of industrial applications. For example, in the food industry, alginates are used as thickeners and stabilizers; in medicine, as wound healing agents, for removing heavy metals and radionuclides from the body; and in cosmetology, as masks for the care of aging skin. NaAlg slowly absorbs water, turning into a colloidal solution or gel that is not destroyed by freezing, thawing, and heating. A unique property of NaAlg is that it is able to absorb water 300 times more than its own weight. NaAlg is used as a food additive E401, thickener, and stabilizer. More information about the above-mentioned properties of NaAlg and other alginates is available in publications [1,2,3,4,5,6,7,8,9,10].

Alginates and their composites have been actively studied using various physicochemical methods [11,12,13,14,15,16,17,18,19,20]. Analyzing the literature data, we noticed that the X-ray photoelectron spectra (XPS) of NaAlg are very different among different authors. For example, in [21], the authors described the C1s spectrum of NaAlg deposited on sphalerite as two peaks. The authors attributed one peak with E_b_ = 288.17 eV to the carbon of the -COONa group, and the second with E_b_ = 284.35 eV to carbon in the ring structure of NaAlg. In [11], it was shown that after treating apatite with NaAlg, in addition to three peaks of random contamination with carbon, two new peaks appeared in the spectrum at 286.92 eV and 288.01 eV, which the authors attributed to the carbon C-O and C=O groups of the polymer. The authors of [15] identified three peaks in the C1s spectrum at 285.46 eV (O-C-O), 283.87 eV (C-O) and 282.20 eV (C-C), respectively. We wanted to understand the reasons for the appearance of such different C1s XPS spectra in position and shape for a thermally stable compound such as NaAlg.

As noted above, NaAlg is a biodegradable polymer. However, at present, it seems to us not entirely correct to associate the differences in the XPS spectra described above with different degrees of polymer degradation. Unfortunately, we did not find any work on the destruction of NaAlg under irradiation with X-ray. However, we found several publications devoted to the interaction of alginates with ^60^Co gamma rays (see [22,23,24,25] and references to them). As is known, during the decay of ^60^Co, two gamma quanta with energies of 1.17 MeV and 1.33 MeV are emitted. Given the obvious difference in the ionizing ability of X-ray and gamma quanta, we considered it possible to present here the main results obtained from studying the degradation of alginates irradiated with ^60^Co radiation.

It was found that during irradiation the average molecular weight of alginates decreases, and this process occurs more efficiently in solution than in the solid state. The degradation curve of alginate irradiated in air was almost the same as that in a vacuum. Degradation is accompanied by a color change from light to dark brown in the case of highly degraded alginate. In the UV spectra of colored alginates, there is a peak at 265 nm, which is associated with the formation of a double bond in the pyranose ring. The nuclear magnetic resonance (NMR) spectra indicated both the formation of carbonyl groups and the formation of double bonds. The analysis of X-ray diffraction patterns showed amorphization of solid alginates during γ-irradiation.

We conducted the first study of the behavior of NaAlg under conditions of long-term X-ray irradiation under ultra-high vacuum conditions. It was found that NaAlg degradation occurs in this case, accompanied by a significant decrease in the O/C ratio. When analyzing the dependence of the intensities of individual peaks in the C1s spectrum on the irradiation time, it was found that after 100 min of irradiation, a peak appears in the spectrum, which is caused by the carbonate group. In the IR spectrum of the NaAlg film, which was irradiated for 300 min, an absorption band was detected at 1910 cm^−1^, which is usually associated with the presence of allene groups. We believe that the observations we described indicate new possible degradation products of alginates when they are irradiated with soft X-ray quanta in a high vacuum, and will also help other authors obtain reliable data when studying NaAlg, its composites and similar compounds using X-ray photoelectron spectroscopy.

## 2. Materials and Methods

NaAlg powder was purchased from AppliChem GmbH - Darmstadt, Germany. Samples for studying XPS spectra were prepared by applying the powder to conductive double-sided tape. In addition, NaAlg films were also prepared for the study. For this purpose, NaAlg powder was kept in water for several days, and then the resulting hydrogel was applied to a glass substrate and dried in air at room temperature (23 °C) for 24 h. The spectra were recorded from the smooth surface of the film, which was initially in contact with the surface of the glass substrate. To study the XPS spectra, the film was mechanically mounted on a sample holder, which was a molybdenum plate.

To record the spectra, we used a Specs PHOIBOS 150 MCD9 electronic spectrometer (Specs, Berlin, Germany), equipped with an X-ray tube with an Mg/Al tween anode (Mg Kα radiation 1253.6 eV, Al Kα radiation 1486.6 eV). When taking spectra, the base vacuum in the spectrometer chamber was 3 × 10^−10^ mbar. The source power was 150 W. The spectra were recorded in the constant transmission energy mode (40 eV for the survey spectrum and 10 eV for individual lines). The survey spectrum was recorded with a step of 1 eV and the spectra of individual lines were recorded with steps of 0.05 eV. Data exported from the instrument-native file format to ISO 14976 [26] (VAMAS) format were analyzed using CasaXPS software (Version 2.3.23 rev. 1.1 R). The values of the relative sensitivity factors that were used in calculating the elemental composition of the polymer are presented in Table 1.

The IR spectra obtained with a resolution of 4 cm^−1^ and 32 scans were recorded at room temperature in the range of 450–4000 cm^−1^ on a Perkin-Elmer “Spectrum Two” FT-IR spectrometer (Waltham, MA, USA) with an ATR attachment.

The microstructure of the film’s surface was studied using a scanning electron microscope (SEM) COXEM EM-30 (Mini SEM; COXEM, Daejeon, Republic of Korea) in backscattered electron imaging at the accelerating voltage of 15 kV. The pressure in the microscope chamber did not exceed 2 × 10^−5^ Pa. For SEM, selected fragments of samples were mounted on aluminum stubs using double-sided adhesive carbon tape (Nisshin EM, Tokyo, Japan).

## 3. Results and Discussion

### 3.1. IR Spectra of NaAlg

Figure 1 shows the IR spectra of NaAlg powder and dry film. It can be seen that they are almost identical. Moreover, they coincide with those given earlier in the literature [12,13,27]. In accordance with the literature data, a broad peak with a maximum at 3223 cm^−1^ is due to stretching vibrations of O-H bonds. The absorption bands around 1596 cm^−1^ and around 1410 cm^−1^ are due to ν_as_(COO) and ν_s_(COO), respectively [21]. The peak with a maximum at 1027 cm^−1^ has a rather complex structure and is attributed to vibration. In the widely cited work [12], all peaks with maxima in the range of 1081–1027 cm^−1^ are attributed to vibrations of the C-O-C group. In [11], the peak at 1021 cm^−1^ was assigned to the stretching vibrations of this group. However, it seems to us that the assignment made in [28] is more correct, in which the peak near 1030 cm^−1^ is associated with OH bending, and the peak near 1080 cm^−1^ with C-O-C stretching.

The intensity of the XPS spectra from the smooth surface of the film significantly exceeds the intensity from the rough surface of the powder preparation (due to shadow effects) at the same X-ray tube power. Consequently, a high-quality spectrum can be obtained from the film in less time. The task of IR spectroscopy, in this case, is to show that during the transition from the powder preparation to the film, no changes occurred in the structure of the sample under study.

### 3.2. XPS Spectra of NaAlg—Dependence on the Choice of Anode

In Figure 2, one can see survey XPS spectra of the NaAlg film obtained using aluminum and magnesium anodes. Table 2 shows the positions of the analytical peaks in the survey spectra. It can be seen that with both excitation methods, the position of the analytical peaks is shifted towards higher bond energies relative to those presented in publications [11,15,21]. We associate these shifts with the charging effect, which is because the film under study is a dielectric material.

From Figure 2, it is clear that when the spectrum is excited by Al Kα quanta, the Na KLL Auger line overlaps with the O1s peak, and when the spectrum is excited by Mg Kα quanta, it overlaps with the C1s peak. Consequently, during the elemental analysis of the composition of the polymer surface, in the first case, an error is possible in determining the integral intensity of the O1s peak, and in the second case, in determining the integral intensity of the C1s peak.

As is known, the shape of Auger spectra does not depend on the method of formation of the primary hole. Confirmation of this can be seen in Figure 3, which shows the O KVV Auger spectra of NaAlg excited by Al Kα and Mg Kα quanta. The different designations for the Auger spectra of sodium and oxygen were chosen deliberately since, in the case of sodium, the valence electron is the 3s (or M_1_) electron, and in the case of oxygen, the valence electrons are 2s and 2p (L_1_ and L_2,3_).

Let us now analyze the Auger spectra of Na KLL NaAlg obtained using different anodes (Figure 4). In the case of an aluminum anode, the O1s peak almost completely overlaps with the Na KL_1_L_2,3_ peak (final state 2s^1^2p^5^ (^1^P_1_)). Consequently, when determining the integrated intensity of the O1s peak using the standard method, in this case we may obtain an overestimated value. In the case of a magnesium anode, the C1s peak is practically not distorted. The contribution of the Na KL_1_L_2,3_ peak intensity to the O1s peak intensity in the case of an aluminum anode can be taken into account according to Equation (1):I^c^(O1s) = I^m^(O1s) − k × I(Na KL_2,3_L_2,3_),(1)
where I^c^(O1s) and I^m^(O1s) are the corrected and measured intensities in the spectrum obtained using an aluminum anode. The value of k is easy to calculate (Equation (2)) from the spectrum that we excited with a magnesium anode:k = I(Na KL_1_L_2,3_)/I(Na KL_2,3_L_2,3_) = 0.251.(2)

### 3.3. Dependence of Spectra on Sample Preparation Method

The intensity of the spectrum from a smooth film is almost two times higher than that from a powder (see the Y scale in Figure 5). This is the expected effect of the sample preparation method. Much more interesting is the dependence of the spectrum shape on the method of sample preparation. To describe a high-energy resolution spectrum in the region of the C1s line, four sum Gaussian/Lorentz (SGL) peaks are required. SGL is calculated using Equation (3):SGL(x) = ηG(x) + (1 −η)L(x),(3)
where G(x)—Gaussian function, L(x)—Lorentz function, η *=* 0.3 in our case.

A similar description of the C1s line of NaAlg was presented in [15,29]. It can be seen that the carbon from the substrate (double-sided tape), on which the powder is attached, distorts the C1s spectrum towards an overestimation of the peak, which is designated as C-C in the figure.

From this observation, it follows that the method of sample preparation may be one of the reasons for the appearance in publications of C1s XPS spectra of NaAlg that differ in position and shape, as reported in the introduction. When studying powder preparations, it is also necessary to remember that the powder is not always densely applied to the support. You can control the possible contribution of the substrate quite easily if you use a substrate material that does not contain elements that are present in the composition of the analyzed sample. In our case, choosing a molybdenum substrate to secure the alginate film seems to be a good choice. There are no molybdenum peaks in the survey spectra, which means that the film under study is quite thick and has no breaks.

### 3.4. Degradation of NaAlg Films

When recording XPS spectra of NaAlg, it was noticed that the shape of the C1s line changes with time. We assumed that perhaps this effect is associated with the destruction of the sample as a result of irradiation with X-ray quanta. We decided to study in more detail the dependence of changes in spectra on irradiation time, up to times significantly exceeding those usually used when recording spectra.

Figure 5 shows C1s spectra obtained after irradiation of a sodium alginate powder and film with X-ray quanta for 5 and 60 min, respectively (characteristic times of the beginning and end of recording XPS spectra for one sample). It can be seen that the shape of the spectrum is changing. The results of the decomposition of the spectra indicate that as a result of irradiation, the intensity of the C-C peak decreases and the intensity of the C-O peak increases. It was interesting to see what happened with longer irradiation times. Let us note here that it is known from the literature that some polymers degrade under the conditions of recording XPS spectra [30,31]. From work [31], it follows that noticeable transformations occur at irradiation times of 200–700 min. In our work, we decided to limit the irradiation time to 500 min. 

Figure 6 shows the dependence of the ratio of elements in the surface layer of sodium alginate on irradiation time. It can be seen that the O/C ratio decreases with time. In other words, the surface of the polymer is reduced during irradiation. The Na/C ratio, on the contrary, increases during the acquisition of spectra. It is obvious that a decrease in the O/C ratio can occur when small molecules such as O_2_, H_2_O, and CO_2_ are released into the gas phase. However, an increase in the Na/C ratio is possible only with the release of CO_2_.

Comparing the shapes of the C1s line before and after five hours of irradiation (Figure 7), one can see that during irradiation the proportion of carbon atoms with a neutral charge increases significantly (peak 1 or peak C-C). In this case, the concentration of carbon atoms having one bond or two bonds with oxygen atoms (peaks 2 and 3) decreases. What was unexpected was the appearance of a new peak in the spectrum of the irradiated polymer, which, based on its position, can be attributed to the carbon atom in the carbonate group.

It was interesting to see how the intensities of individual peaks in the C1s spectrum changed during irradiation. The results of the decomposition of the spectra at each time point are presented in Figure 8 and Table 3. From the presented graph, it can be seen that peak 5 (carbonate group) appears only after 100 min of irradiation.

Two new peaks (2638 and 1910 cm^−1^) appear in the IR spectrum of the film irradiated for 300 min (Figure 9). The peak at 2638 cm^−1^ can be associated with carboxylic acid, and the peak at 1910 cm^−1^, according to [32], is clearly due to allene groups. The appearance of an inflection in a broad absorption band located in the range from 1800 to 1500 cm^−1^ can be described by the appearance of a new peak at 1707 cm^−1^ (peak 3 in the inset), which can be attributed to carbonate groups. The latter is in accordance with the conclusion drawn from the analysis of X-ray photoelectron spectra. Nevertheless, it should be noted here that the authors of [33] previously observed a peak at 1709 cm^−1^ in the spectrum of irradiated NaAlg. They associated its appearance with the stretching vibrations of the C=O ketone group. It is obvious that the contribution to the intensity of broad peak 2 in the insert (1639 cm^−1^) can be made by stretching vibrations of alkene groups.

SEM images of the surface of NaAlg films before and after irradiation are presented in Figure 10. Initially, NaAlg films had an appearance characterized by cleanliness, smoothness and uniformity over the entire surface. However, after exposure to X-ray radiation, a noticeable transformation of the film surface occurred. On the surface, one can see primarily deep cracks, small peeling areas and bulges. All this can be considered signs of destruction of the original structure of NaAlg.

In particular, visually, after 500 min of irradiation, the film ceased to be transparent and its color acquired a yellow-brown tint. The film could not be completely separated from the double-sided adhesive carbon tape using tweezers and a scalpel; it became more fragile. These changes in physical properties indicate significant damage and degradation that has occurred under X-ray exposure.

As noted in the introduction, NaAlg is a biodegradable polymer. However, it currently seems to us not entirely correct to compare the above-described changes in the structure of the polymer under the influence of soft X-ray quanta with changes resulting from the influence of chemical reagents and/or biological factors [15,23,24,25]. Unfortunately, we did not find any other work on the destruction of NaAlg under irradiation with X-ray quanta; however, we found a large number of works devoted to the interaction of alginates with ^60^Co gamma rays [25]. As is known, during the decay of ^60^Co, two gamma quanta with energies of 1.17 MeV and 1.33 MeV are emitted. Given the obvious difference in the ionizing ability of X-ray and gamma quanta, we considered it possible to present here the main results obtained from studying the degradation of alginates irradiated with ^60^Co gamma rays.

It has been observed that the average molecular weight of alginates decreases during irradiation, with this process being more pronounced in solution compared to the solid state. Interestingly, the degradation curve of alginate irradiated in air closely resembles that of alginate irradiated in a vacuum environment. As degradation progresses, highly degraded alginate exhibits a color transformation from a light to a dark brown hue. The ultraviolet (UV) spectra of the colored alginates reveal a distinct peak at 265 nm, indicating the formation of a double bond in the pyranose ring.

Further analysis using NMR spectroscopy confirmed the presence of carbonyl groups and double bonds as byproducts of the degradation process. Additionally, X-ray diffraction patterns demonstrated the amorphization of solid alginates under gamma ray irradiation. These findings suggest that the changes induced by X-ray irradiation on NaAlg align with the previously observed results.

Moreover, the visual manifestation of color alteration in alginate remains consistent across both types of irradiation. Our data also point towards redox processes occurring during the degradation of NaAlg, as evidenced by the analysis of XPS spectra. Notably, we have identified the formation of carbonate and allene groups as novel products resulting from irradiation. Carbonate groups may serve as end groups upon polymer chain breakage, while allene groups are generated through the liberation of water molecules into the gas phase. This study sheds light on the intricate chemical transformations that NaAlg undergoes under X-ray irradiation, offering new insights into its degradation mechanisms.

The changes that occur with NaAlg during irradiation with X-ray quanta do not contradict the described results. The visual description of the color change in alginate is simply very similar for both types of irradiation. Our data also indicate that degradation is accompanied by redox processes. For the first time, we indicate the formation of carbonate and allene groups upon irradiation. Possible reaction schemes are presented in Figure 11 in accordance with works [24,25,33].

The Lebedev intramolecular reaction and Zaitsev’s rule play crucial roles in the formation of double bonds within allene groups. According to these principles, it is expected that predominantly double bonds of allene groups will be formed, as this configuration is the most thermodynamically favorable. This insight is particularly important in understanding the chemical transformations that occur during the degradation of NaAlg films.

In the proposed scheme for the degradation reactions of NaAlg films, the formation of carbonate carbon is assumed to occur through Ruzicka cyclization. This process involves the intramolecular cyclization of a linear precursor to form a cyclic structure. This step is significant in elucidating the overall degradation pathway of NaAlg films. The chemical reactions involved in this degradation process can be described in detail. Initially, the excited fragments of the alginate molecule are formed directly by irradiation. Subsequently, the radical of the alginate molecule (R▪) is generated by the extraction of H▪ from RH∗ [33].

The resulting R▪ radicals are unstable due to their strained structures. Consequently, they decompose with the cleavage of glycosidic bonds to produce 1a and 1b before scission with the loss of water into the gas phase (1c). Radical 1a leads to the formation of 2a, which contains a carboxyl group. Subsequently, 2a undergoes intramolecular cyclization and shifts to form 3b and NaHCO_3_. The carboxyl group at 1c, while 2b is dehydrogenated by H▪ radical or high molecular weight radical (R▪), induces the formation of unsaturated compounds with double bonds at 1c and 2c with the loss of water.

The scheme presented in Figure 11 provides a visual representation of these intricate chemical transformations, offering valuable insight into the degradation pathways of NaAlg films.

## 4. Conclusions

Thus, the changes that occur with NaAlg during irradiation with X-rays do not contradict the results described in the introduction on the study of degradation during irradiation with gamma rays, namely:(1)Degradation is accompanied by a change in the color of NaAlg from light to yellow-brown;(2)NaAlg is amorphized as a result of irradiation;(3)When irradiated, the average molecular weight of alginates decreases;(4)It has been established that during the process of radiation degradation, the formation of both carbonyl groups and double bonds occurs.

The visual description of the color change in alginate is very similar for both gamma- and X-ray types of irradiation. Our data also indicate that redox processes accompany the degradation of NaAlg under the conditions of the recording of XPS spectra. For the first time, we indicate the formation of carbonate and allene groups upon X-ray irradiation. Note that carbonate groups can act as end groups when the polymer chain breaks and allene groups are formed because of the release of water molecules into the gas phase.

In conclusion, we can probably repeat that, as established in this work, NaAlg noticeably degrades when irradiated with soft X-ray quanta under ultra-high vacuum conditions; therefore: A) when taking its XPS spectra, long exposures should not be used; B) to preserve the beneficial properties of NaAlg, contact with sources of ionizing radiation should be avoided.

## Figures and Tables

**Figure 1 polymers-16-02072-f001:**
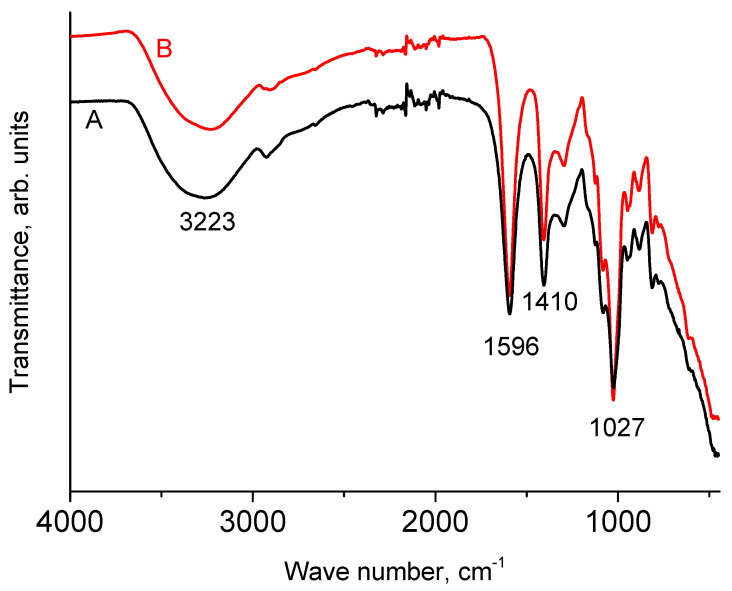
IR spectra of powder (A) and film (B) of NaAlg.

**Figure 2 polymers-16-02072-f002:**
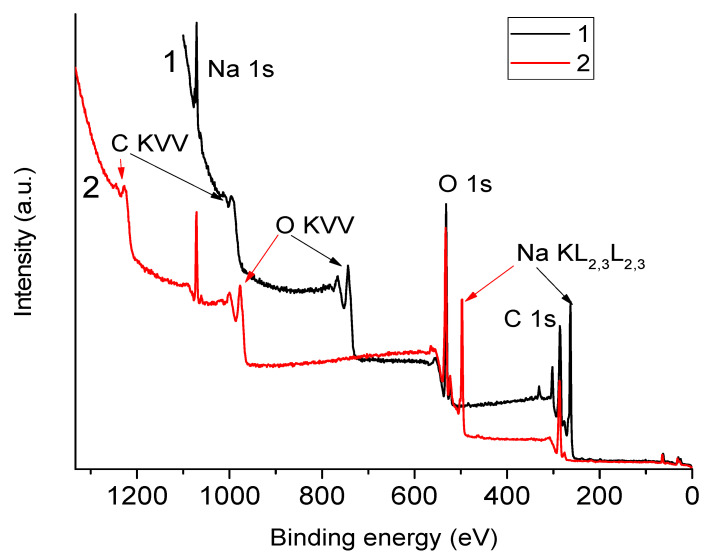
Survey XPS spectra of a NaAlg film excited using magnesium (1) and aluminum (2) anodes.

**Figure 3 polymers-16-02072-f003:**
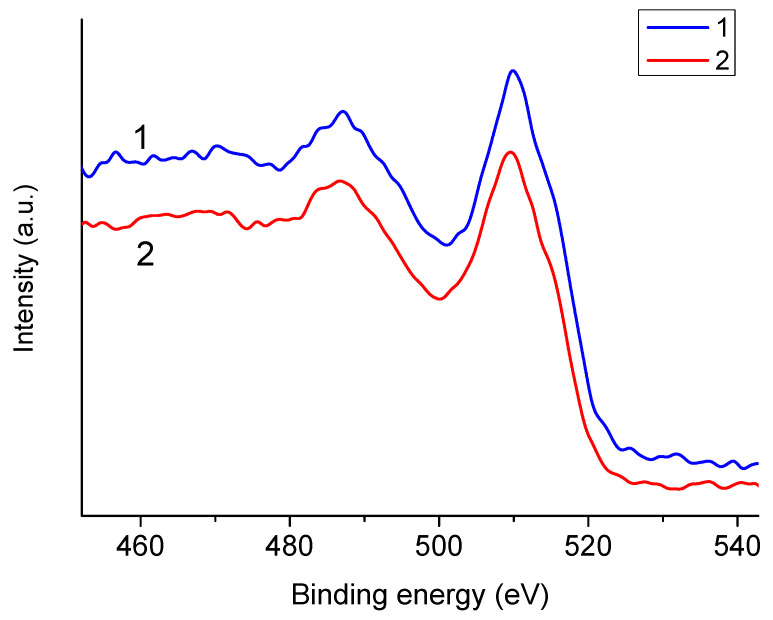
O KVV Auger spectra of NaAlg excited using aluminum (1) and magnesium (2) anodes.

**Figure 4 polymers-16-02072-f004:**
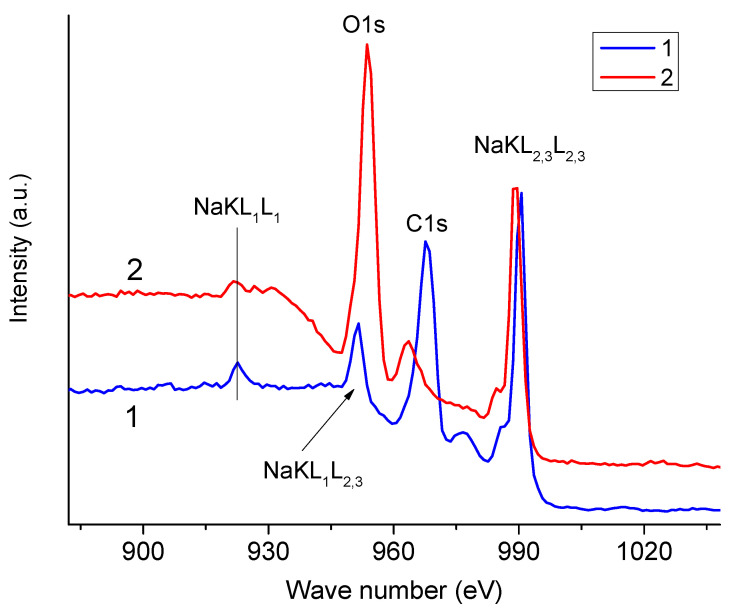
Auger spectra of Na KLL NaAlg excited using aluminum and magnesium anodes.

**Figure 5 polymers-16-02072-f005:**
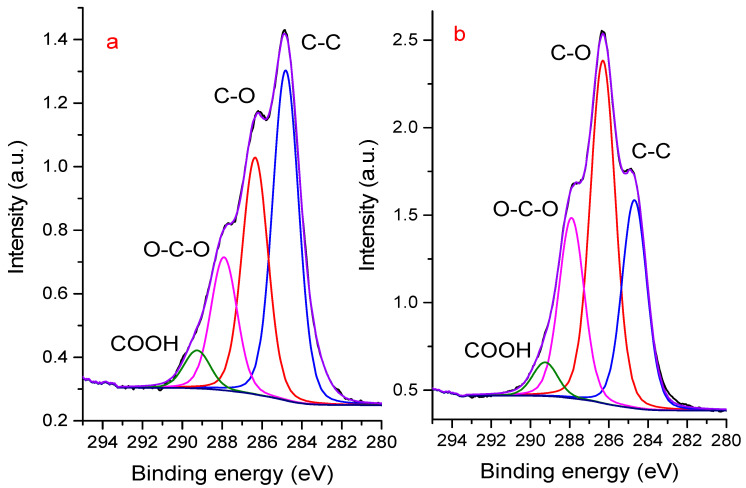
XPS spectra of C 1s powder (**a**) and film (**b**) of sodium alginate excited using an aluminum anode after 5 (**a**) and 60 (**b**) minutes of irradiation X-ray. The decomposition of the spectra and the assignment of individual peaks were carried out by analogy with work [15].

**Figure 6 polymers-16-02072-f006:**
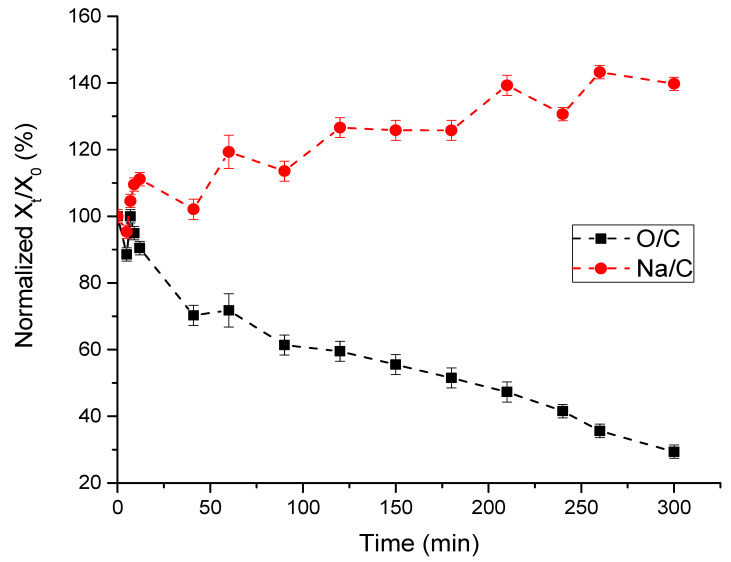
Changes in the ratio of O/C and Na/C in the surface layers of sodium alginate film during its irradiation with rays of an X-ray tube with an Mg anode.

**Figure 7 polymers-16-02072-f007:**
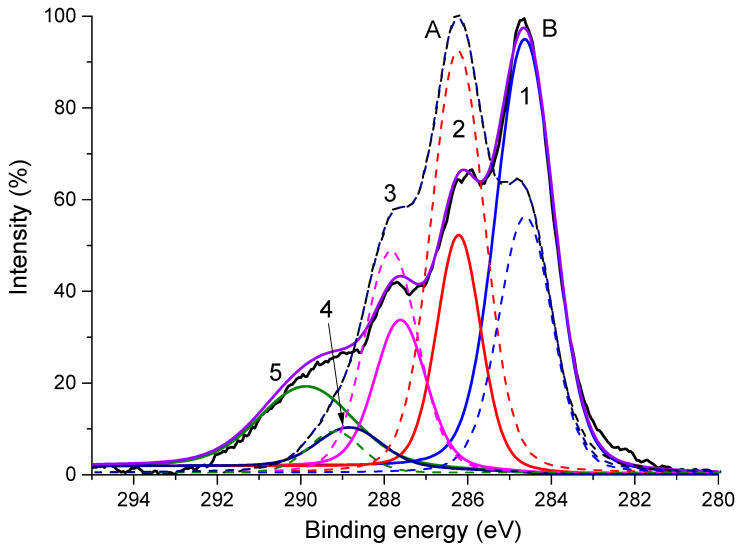
C1s spectra of sodium alginate film before irradiation (spectrum A, dotted lines) and after Mg Kα X-ray irradiation for 5 h (spectrum B, solid lines).

**Figure 8 polymers-16-02072-f008:**
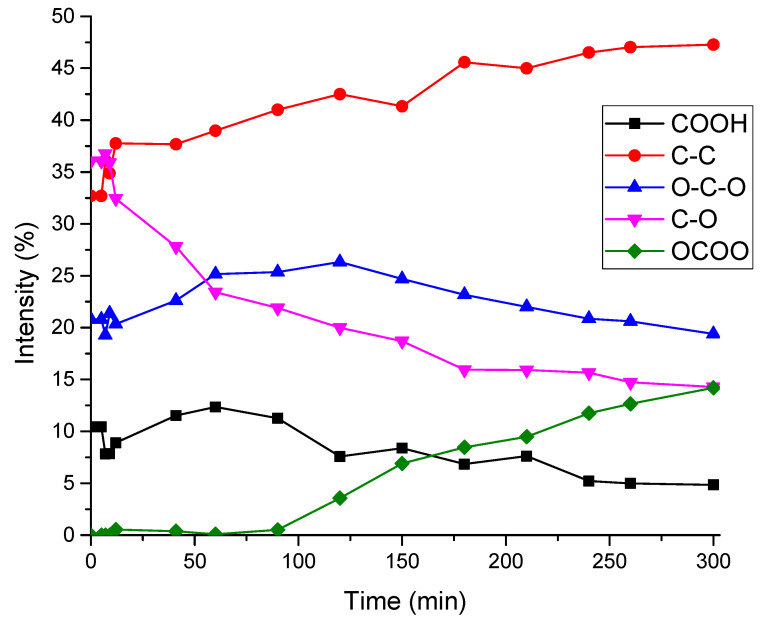
Dependences of the relative intensities of individual peaks in the C1s spectrum on irradiation time. The total intensity of all peaks at each time point is 100%.

**Figure 9 polymers-16-02072-f009:**
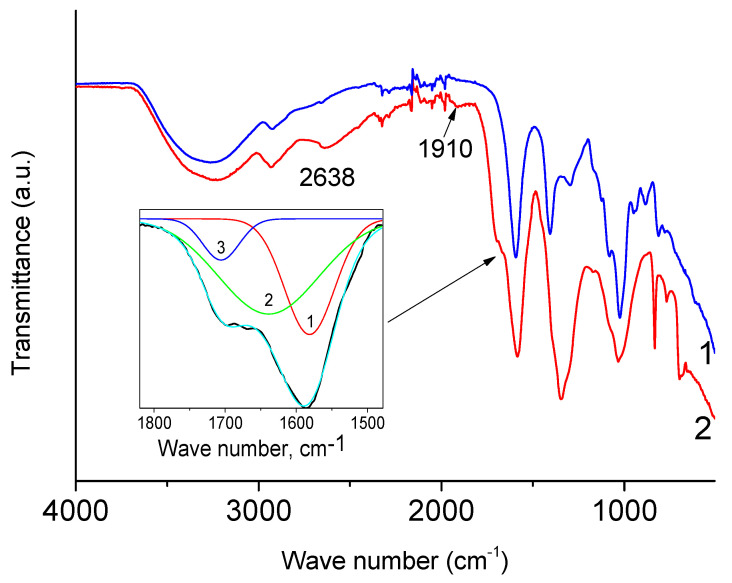
IR spectra of sodium alginate film before (1, blue line) and after (2, red line) irradiation. The inset shows the decomposition of a fragment of the IR spectrum of the irradiated film in the range of 1800–1500 cm^−1^.

**Figure 10 polymers-16-02072-f010:**
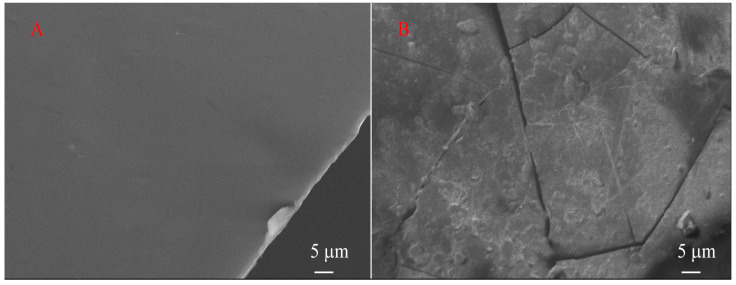
SEM of NaAlg film before (**A**) and after (**B**) Mg Kα X-ray irradiation.

**Figure 11 polymers-16-02072-f011:**
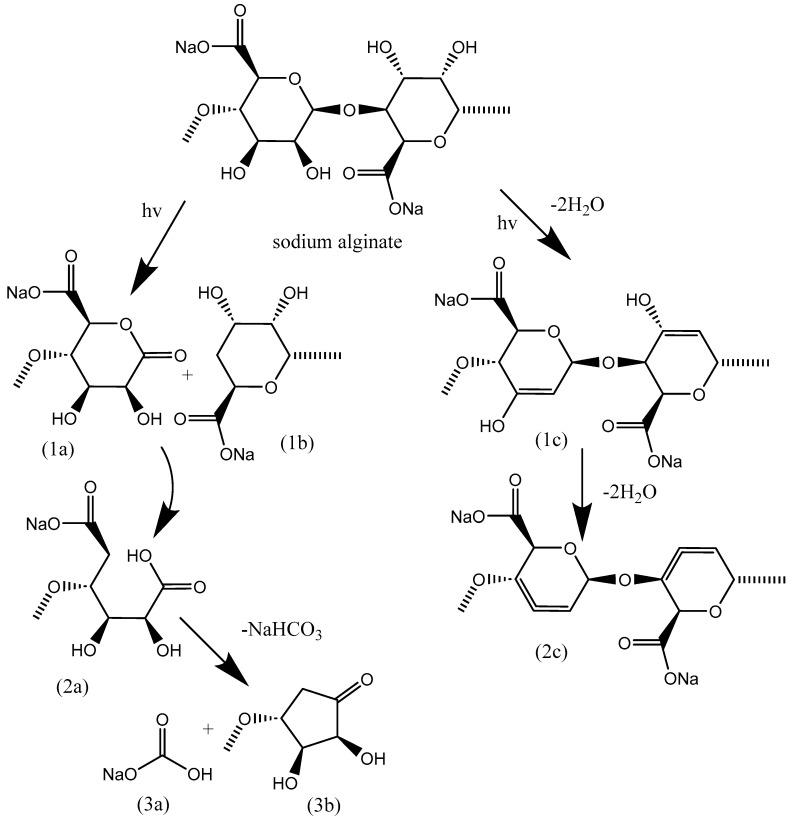
Scheme of the proposed degradation reactions of NaAlg films under X-ray irradiation.

**Table 1 polymers-16-02072-t001:** Relative sensitivity factors (R.S.F.).

Anode	O1s	C1s	Na1s
Al	2.93	1	8.52
Mg	2.85	1	7.99

**Table 2 polymers-16-02072-t002:** Positions and half-widths of analytical peaks in the survey XPS spectra of the NaAlg film.

Anode	Peak	Binding Energy, eV	FWHM, eV
Al	Na1s	1071	3
	O1s	533	3.9
	C1s	287	4.5
Mg	Na1s	1071	2.5
	O1s	532	3.3
	C1s	286	4.4

**Table 3 polymers-16-02072-t003:** Position (in eV) and intensity (in %, in parentheses) of the peaks in the C1s spectra are presented in Figure 7.

Peak Number	1	2	3	4	5
Assignment	C-C	C-O	O-C-O/C=O	COOH	CO_3_^2−^
Spectrum A, eV (%)	284.9 (47.27)	286.5 (14.29)	287.9 (19.39)	289.2 (4.85)	290.2 (14.2)
Spectrum B, eV (%)	284.7 (32.69)	286.3 (36.06)	287.9 (20.8)	289.2 (10.44)	—

## Data Availability

The original contributions presented in the study are included in the article, further inquiries can be directed to the corresponding authors.

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
