# Peer review of "Degradation of Polymer Films of Sodium Alginate during Prolonged Irradiation with X-ray under Ultra-High Vacuum"

_polymers, 2024, doi:10.3390/polym16142072_

Round 1

Reviewer 1 Report (Previous Reviewer 5)

Comments and Suggestions for Authors

The authors had the opportunity to improve their submission. However, there is no extra analysis to improve the study.

It is unknown if the degradation of alginate due to the vacuum or the XPS radiation.

The time of the film exposure is long and unusual.

Comments on the Quality of English Language

Language should be improved.

Author Response

Reviewer 2 Report (New Reviewer)

Comments and Suggestions for Authors

The manuscript "polymers-3082723" by Kabachkov et al. reported the Degradation of sodium alginate during the recording of XPS spectra. After review, this study is interesting. It can be said that new observations are reported in this manuscript. But the authors have to make minor changes. The authors should refer to the following comments to improve their work:

1. Please rewrite the abstract. Use more content.

2. Please improve the discussion.

3. You don't need to show Figure 1 as A and B. Please report in one form.

4. The terms X-ray, NMR, IR, SEM, XPS, and UV are abbreviated in the manuscript without explanation, please revise and rewrite. After explaining an abbreviation for the first time, use it later.

5. If you introduce an abbreviation for sodium alginate, use the abbreviation in the manuscript.

6. The language of the manuscript should be checked.

Comments on the Quality of English Language

Minor editing of English language required.

Author Response

Reviewer 3 Report (New Reviewer)

Comments and Suggestions for Authors

The study proposes a mechanism for Na Alg degradation under X-ray irradiation. The proposed mechanism is supported by XPS and IR data. The study is of interest for the readers of Polymer journal, the manuscript is well written and the data are correctly discussed.

My comments are:

-          Paragraph 3.4. Degradation of sodium alginate films:  In fig 5 the authors compare the C1s spectra of powder and film of Na Alg. The figure with C1s spectra of film after 5 and 60 minutes of irradiation is missing.

-          Please correct the caption of figure 7, in the text spectrum A (solid line ) is referred to irradiated sample and spectrum B (dotted line) is referred to the sample before the irradiation.

Author Response

Reviewer 4 Report (New Reviewer)

Comments and Suggestions for Authors

In this manuscript, the Authors presented results of degradation of sodium alginate, powder, and films, upon prolonged irradiation with X-ray under high vacuum conditions. The Authors analyzed XPS and FTIR spectra and indicated the formation of carbonate and allene groups upon X-ray irradiation.

The idea of the work is interesting, but the experiment and results are not presented enough. I don't think the title is appropriate either. From the context of the work, it is more appropriate: Degradation of sodium alginate under X-ray exposure: XPS and FTIR spectra study.

Here are some suggestions for the text improvement:

1.     The abstract should be written more precisely and informatively. It should present the main outcome of the paper while stating the main hypothesis behind the work. The text of the paper compares the spectra of NaAlg powder and films, which are not visible in the abstract. Additional suggestions are indicated in the form of comments in the pdf version of the manuscript.

2.     The introduction should be written systematically and concisely, without repeating facts. The main novelty in this work must be pointed out in the introduction.

3.     It is unusual for assumptions, made based on presented results, to be made within the materials and methods section

4.     The image title should be on the same page as the image

5.     Show the spectra in Figures 1a and 1b on the same image.

6.     In section 3.4:

·       It is not clear what changes in the XPS spectrum the authors mean when the recording time is 1 to 60 min.

·       How characteristic is the chosen radiation time of 500 min for the experimental setup, in Figure 7 are the spectra obtained after 5 hours....

·       To specify why the XPS spectra obtained using aluminum anode are given in Figure 5, and then the other images (6 and 7) are given for magnesium anode, without indicating in the discussion itself.

7.     The conclusion should summarize the main findings and clarify how the results presented in the paper have made the field progress. A repetition and listing of the results for a summary of the findings is not sufficient. 

Round 2

Reviewer 1 Report (Previous Reviewer 5)

Comments and Suggestions for Authors

The authors addressed most of the comments and the revised version can be accepted.

Comments on the Quality of English Language

Professional language editing should be conducted.

Reviewer 4 Report (New Reviewer)

Comments and Suggestions for Authors

The manuscript can be accepted for publication in its present form.

This manuscript is a resubmission of an earlier submission. The following is a list of the peer review reports and author responses from that submission.

Round 1

Reviewer 1 Report

Comments and Suggestions for Authors

Only degradation study is not suitable to defend a research idea. It could have been related to some application or activity of the sodium alginate. Like in the following study the author  has correlated oxidation with degradation and gelation of solium alginate

A Study on the Correlation between the Oxidation Degree of Oxidized Sodium Alginate on Its Degradability and Gelation. Polymers (Basel). 2022 May; 14(9): 1679.

Overall the study is very short, more experimentation may be perfomed to correlating degradation with other properties 

Comments on the Quality of English Language

satisfactory

Reviewer 2 Report

Comments and Suggestions for Authors

The paper is very good work, but I have two requests before its acceptance.

First, please, provide more detailed SEM observations of the morphology.

Second, to provide spectroscopic analysis to verify the contamination of the fiber surface by matrix residues.

Reviewer 3 Report

Comments and Suggestions for Authors

Reviewer 4 Report

Comments and Suggestions for Authors

The authors undertook the study of the behavior of NaAlg under conditions of recording XPS spectra. It was also found that during prolonged irradiation with X-ray under high vacuum conditions, degradation of NaAlg occurs, which is accompanied by a significant decrease in the O/C ratio. This is a very interesting work. However, there are same comments should be addressed;

1. Why provide infrared spectra of NaAlg powder and dry film? It needs to be explained in the text.

2. It is recommended to add an introduction to XPS in the introduction.

3. The formulas in this paper need to be added to the corresponding references.

4. Does CO2 in the air affect its carbon and oxygen analysis? How to eliminate its effects.

5. Error estimates should be included throughout. In addition, the number of repeat experiments underlying error estimates should be specified throughout in the respective figure/table caption

6. The topic is the use of XPS technology, so it is necessary to summarize and summarize the general situation of this technology. Is it just for sodium alginate?

Reviewer 5 Report

Comments and Suggestions for Authors

The authors reported the degradation of sodium alginate during the recording of XPS spectra. The submission is unacceptable for the following points:-

1.       The title should be revised to be informative reflecting the authors’s main findings. Remove abbreviations.

2.       The novelty of the study is low and should be improved.

3.        There is no sufficient motivation for the study.

4.       The measurements were investigated for a very long time with no explanation.

5.       The product of degradation should be fully characterized.

6.       The mechanism of degradation should be proposed.

7.       The author’s point of view to avoid degradation should be added.

8.       The language should be revised and typos should be corrected.

Comments on the Quality of English Language

Professional language service should be conducted.

Round 2

Reviewer 2 Report

Comments and Suggestions for Authors

For me, could be accepted

Comments on the Quality of English Language

Check some small grammar errors